# FM1-43 Dye Memorizes Piezo1 Activation in the Trigeminal Nociceptive System Implicated in Migraine Pain

**DOI:** 10.3390/ijms24021688

**Published:** 2023-01-14

**Authors:** Adriana Della Pietra, Nikita Mikhailov, Rashid Giniatullin

**Affiliations:** A.I. Virtanen Institute for Molecular Sciences, University of Eastern Finland, 70211 Kuopio, Finland

**Keywords:** pain, migraine, nociception, Piezo1, trigeminal neuron, glia, FM1-43

## Abstract

It has been proposed that mechanosensitive Piezo1 channels trigger migraine pain in trigeminal nociceptive neurons, but the mechanosensitivity of satellite glial cells (SGCs) supporting neuronal sensitization has not been tested before. Moreover, tools to monitor previous Piezo1 activation are not available. Therefore, by using live calcium imaging with Fluo-4 AM and labeling with FM1-43 dye, we explored a new strategy to identify Piezo channels’ activity in mouse trigeminal neurons, SGCs, and isolated meninges. The specific Piezo1 agonist Yoda1 induced calcium transients in both neurons and SGCs, suggesting the functional expression of Piezo1 channels in both types of cells. In Piezo1-transfected HEK cells, FM1-43 produced only a transient fluorescent response, whereas co-application with Yoda1 provided higher transient signals and a remarkable long-lasting FM1-43 ‘tail response’. A similar Piezo1-related FM1-43 trapping was observed in neurons and SGCs. The non-specific Piezo channel blocker, Gadolinium, inhibited the transient peak, confirming the involvement of Piezo1 receptors. Finally, FM1-43 labeling demonstrated previous activity in meningeal tissues 3.5 h after Yoda1 washout. Our data indicated that trigeminal neurons and SGCs express functional Piezo channels, and their activation provides sustained labeling with FM1-43. This long-lasting labelling can be used to monitor the ongoing and previous activation of Piezo1 channels in the trigeminal nociceptive system, which is implicated in migraine pain.

## 1. Introduction

Despite the high prevalence of migraines in the world’s population [1,2,3], its molecular mechanisms remain poorly understood. However, growing evidence suggests that the trigeminovascular system, composed of trigeminal neurons and meningeal vessels, is the leading actor in the generation of migraine pain [4]. We previously proposed that trigeminal neurons surrounded by pulsating vessels trigger the nociceptive cascade via activation of mechanosensitive Piezo1 channels [5,6]. Moreover, apart from neurons and vessels, glial cells also contribute to chronic pain conditions [7]. In particular, satellite glial cells (SGCs) can crosstalk with neurons and modulate neuronal activity, promoting neuroinflammation [8,9]. However, the non-neuronal glial component has not been considered in previous studies on Piezo1 channels in meningeal nociception. It is still unclear whether glial cells even express mechanosensitive Piezo1 channels and whether these receptors contribute to their crosstalk with neurons, leading to neuroinflammation and neuronal sensitization, which are contributing factors to migraine pain. The high calcium permeability of Piezo channels [10] suggests that their activation can initiate a cascade of anti- and pro-inflammatory cytokines [11]. Therefore, the identification of currently and previously activated Piezo1 channels would unleash molecular mechanisms of the inflammatory and pro-nociceptive triggers that contribute to the onset of migraine pain. The detection of Piezo1 mediated mechanisms requires a range of specific tools that are applicable to living cells, ex vivo preparations, and in vivo experiments. 

However, to date, there is a lack of methods for the retrospective detection of Piezo1 activity. The styryl FM1-43 dye, traditionally known as the marker of synaptic vesicle recycling, has also been suggested to label cells due to its penetration through mechanosensitive channels [12,13], but the identity of the channels driving the process remains uncertain. More recently, FM1-43 has been used to mark the tactile sensory innervation of oral mucosa [14]. Interestingly, in this morphological study, the authors also identified the presence of Piezo2 channels in corpuscular endings and Merkel cells, although they did not link the role of these channels in the FM1-43-mediated labeling of mechanosensitive nerve fibers. 

In this project, we used a combination of functional and morphological tests to develop a method for labelling Piezo1 channel activity in the cells of the trigeminovascular system with FM1-43, and we explored presence of the mechanosensitive channels in SGCs. We report that trigeminal SGCs, along with neurons, can be activated by the specific Piezo1 agonist Yoda1 and that FM1-43 can detect this activation in real time. Moreover, the persistent FM1-43 “tail response” could serve as a specific marker of previous Piezo1 activity that can be used to monitor mechanosensitivity within the trigeminal nociceptive system. 

## 2. Results

### 2.1. Trigeminal Neurons and Glia Have Diverse Piezo Channel Activity

As the functional expression of Piezo1 receptors in SGCs has not been addressed before, we first used trigeminal cultures to explore calcium responses induced by a specific agonist of Piezo1 receptors (Yoda1) in different cell types. Trigeminal culture is heterogeneous and contains both neurons and SGCs interacting with each other within the trigeminal nociceptive system [15]. Numerous SGCs surround a few big neurons (Figure 1A). SGCs are distinguishable from the bigger oval/round-shaped neurons by their exceedingly small size and strictly round shape (Figure 1A). Figure 1B shows sample prolonged calcium transient traces that are typical for shape variability in Piezo1 [5,16] in individual trigeminal neurons (*n* = 9/40 responsive neurons) after application of the Piezo1 specific agonist Yoda1 (5 μM). Notably, the SGCs, like neurons, also responded to Yoda1 by the transient elevation of intracellular calcium (*n* = 12/27 responsive SGCs), indicating the functional expression of Piezo1 receptors in this specific type of trigeminal glia (Figure 1C). The histograms in Figure 1D show the relative amplitude of Piezo1-mediated responses in neurons vs. SGCs, which did not significantly differ (*p* = 0.120).

Thus, these data demonstrated the functional expression of mechanosensitive Piezo1 channels in both neurons and relative SGCs.

### 2.2. FM1-43 Traps in Transfected HEK Cells after Piezo1 Channel Activation 

Next, we explored whether the styryl FM1-43 dye can specifically report the activity of Piezo1 channels in trigeminal neurons and SGCs. To this end, we tested whether this dye could internalize after specific activation in the Piezo1-transfected HEK293T cells and remain located inside the cells after agonist washout. 

Figure 2A represents FM1-43 labelling of the transfected HEK cells in various stages of the imaging experiment. The first frame shows the HEK cells in response to the 2 min application of FM1-43 alone, which resulted in pure membrane labelling (Figure 2A). In the second snapshot, the following washout removed almost all the previous staining. However, after the co-application of 25 μM of Yoda1 with FM1-43 (Figure 2A, third frame), the dye was trapped in the cytosol (Figure 2A, fourth frame). Notably, this labeling remained stable for at least several minutes of washout (Figure 2D). 

Figure 2B shows a representative sample trace of the changes in fluorescence intensity in one of the cells treated twice with FM1-43 alone, followed by a combination of FM1-43 and Yoda1. The first two transient fluorescence peaks (Figure 2B) corresponded to the labelling of only the membrane (Figure 2A, first frame). Notably, the fluorescence comes back to baseline when FM1-43 was washed out (Figure 2B). In contrast, the application of FM1-43 and Yoda1 evoked a larger transient peak, which was then followed by the lasting ‘tail response’ (Figure 2B). This ‘tail response’ did not show any decay for up to 5 min (Figure 2B). Thus, these results indicated that FM1-43 remained trapped only in the cells in which the Piezo1 channels were activated.

Figure 2C,D shows the results of the statistical analysis for the transient peak and the tail response produced by the application of FM1-43 plus Yoda1, compared with the previous FM1-43 treatment alone. The percentages of increase for the transient peaks were calculated and normalized to the previous baseline level. The tail responses were calculated and normalized to the initial baseline level. The Piezo1-transfected HEK cells responded with a transient peak increase of 65.7 ± 8.7% (*n* = 25, *p* < 0.001) with the application of FM1-43 alone. During the application of FM1-43 plus Yoda1, the transient peak increased to 154.7 ± 25.3% (*n* = 25, *p* < 0.001) with respect to the previous baseline, which was significantly higher than the application of FM1-43 alone. The tail response, in the case of FM1-43 treatment alone, was at baseline levels, corresponding to 0.9 ± 0.6% (*n* = 25, *p* < 0.001). The combination of FM1-43 + Yoda1 produced a much higher and sustained tail response of 22.3 ± 4.7% (*n* = 25, *p* < 0.001). 

These results indicated that Piezo1 activation promotes sustained FM1-43 trapping in the HEK cells expressing this type of mechanosensitive channel.

### 2.3. FM1-43 Memorizes Piezo1-Mediated Events in the Trigeminal Nociceptive System 

Our next aim was to demonstrate that FM1-43 can provide long-lasting reporting of Piezo1 activation in the trigeminal nociceptive system. Therefore, we used primary mouse trigeminal cultures composed of both neurons and SGCs. 

Figure 3A shows a representative sample trace of the variations in fluorescence intensity for a trigeminal neuron treated with FM1-43 alone or in combination with Yoda1. In this set of experiments, we also tested whether the transient or tail responses promoted by Yoda1 could be modified by the nonspecific inhibitor of Piezo1 channels, Gd^3+^. Comparable results were also obtained in SGCs (Figure 3D). 

In the primary cells (both neurons and SGCs), the application of 2 μM of FM1-43 for 2 min alone did not produce either an essential increase in fluorescence or a ‘tail response’ (Figure 3A,D). However, the combination of FM1-43 plus 25 μM of Yoda1 evoked a large signal followed by a prominent ‘tail response’ (Figure 3A), which resembled the tail responses observed in Piezo1-expressing HEK cells. Notably, this combination with 100 μM of Gd^3+^ largely depressed only a transient response (Figure 3A), verifying the blockage of the membrane Piezo1 channels. However, Gd^3+^ did not affect the ‘tail response’ that was generated previously by internalized FM1-43, which, in this case, lasted up to 10 min with a minimal trend for further decay. Comparable results were also obtained in SGCs (Figure 3D), which were consistent with the functional expression of Piezo1 channels shown in Figure 1. 

Figure 3B shows the results of FM1-43 treatment in neurons regarding the transient peak (193.2 ± 13.8% increase; *n* = 74, *p* < 0.001). Notice that, in this case, the tail response was very small (10.5 ± 3.3% over baseline; *n* = 32, *p* < 0.001) (Figure 3C). In contrast, the combination of Yoda1 with FM1-43 induced a larger transient peak (369.9 ± 36.5%, *n* = 74, *p* < 0.001) and a much bigger tail response (91.4 ± 12.7%, *n* = 32, *p* < 0.001). The transient peak that was induced by a combination of Yoda1 and FM1-43 decreased by Gd^3+^ (130.5 ± 9.7%; *n* = 74). However, the tail response was not affected (89.5 ± 18.6%; *n* = 16, *p* = 0.713; Figure 3B,C). 

In SGCs, the application of FM1-43, similarly to neurons, produced a transient peak (175.5 ± 15.5%; *n* = 81) that was lower than treatment with both FM1-43 and Yoda1 (291.3 ± 21.4%; *n* = 81, *p* < 0.001; Figure 3E,F). The tail response was minimal (12.7 ± 2.5%; *n* = 44) after FM1-43, but Yoda1 applied together with FM1-43 produced a valuable tail response (38.44 ± 5.1%; *n* = 44, *p* < 0.001). Gd^3+^ reduced the transient FM1-43 response to 138.7 ± 9.3% (*n* = 81, *p* < 0.001), but the tail response after Gd^3+^ application, like in neurons, stayed stable (35 ± 6%; *n* = 34, *p* = 0.560; Figure 3E,F). The depressant effect of Gd^3+^ on the transient peaks following treatment with FM1-43 plus Yoda1 supported the earlier finding of the functional expression of Piezo1 in SGCs.

In summary, these experiments in primary cells clearly showed that the FM1-43 dye could be trapped for many minutes inside the cells with previously activated Piezo1 channels.

### 2.4. FM1-43 Detected Several Events in Perivascular Meningeal Areas Ex Vivo

Given that FM1-43 remains trapped after Piezo1 activation, we next tested whether FM1-43 is suitable for the labelling of living tissues, and whether it can remain for a longer period, which is a prerequisite for the testing of Piezo1 channels in vivo. In this case, we used a more reliable model for testing migraine pathology, such as an ex vivo meningeal preparation [17,18]. This selection was based on the emerging view that meningeal tissues, composed by blood and lymphatic vessels, multiple nerve fibers, and local immune cells, are supposed to be the origin site for migraine pain [17,19,20,21]. 

Here we aimed to demonstrate that previous Piezo1 activity results in the persistent FM1-43 labeling of the meninges, which lasts for three hours after withdrawal of the transient 10 min Yoda1 stimulation. To this end, we compared the action of 10 μM of FM1-43 alone (Figure 4A) and the action of FM1-43 plus 25 μM of Yoda1 (Figure 4B), which were co-applied for just 10 min. 

In order to address the whole spectrum of meningeal cells, we tested fluorescence changes in a full image of the meninges (Figure 4C). We found a significant increase in fluorescence in certain structures of the meninges after combined FM1-43 plus Yoda1 stimulation compared to the application of FM1-43 alone (Figure 4C), which was maintained after a 3.5 h washout period (*n* = 7, *p* = 0.002).

Thus, we found that trapping of the specific label for several hours demonstrates the previous activity of Piezo1 channels in the meningeal preparation model. 

## 3. Discussion

The main novel findings of the current study can be summarized in two main points: (i) satellite glial cells (SGCs), that surround trigeminal neurons and support neuroinflammation and neuronal sensitization, express functional Piezo1 receptors and (ii) FM1-43 can report the previous activity of Piezo1 channels through long-lasting labeling of the key components of the trigeminal nociceptive system implicated in migraines, including neurons and SGCs. Given the high calcium permeability of Piezo1 channels, these findings extend the role of mechanosensitive mechanisms from neurons to glial cells, which, in concert, can react to various mechanical (or Piezo1-specific chemicals) stimuli by a calcium-dependent mechanism of migraine-related sensitization. 

Mechanosensitive Piezo1/2 receptors emerged recently as the most sensitive detectors of mechanical stimuli [22]. Piezo1, but not Piezo2, can also be activated by certain chemical agents, such as the specific Piezo1 agonist Yoda1 [23]. These professional mechanosensory Piezo channels appear to be key players in many physiological functions, including pain, touch detection, and the control of blood pressure [24,25]. Based on our findings of the functional expression of mechanosensitive Piezo1 receptors on trigeminal neurons [5], we previously proposed their role in typical migraine symptoms, including the generation of a pulsatile type of pain and mechanical hyperalgesia [6]. 

Here, for the first time, we observed the functional activity of Piezo1 in trigeminal SGCs. This was directly detected by Yoda1-induced calcium transients in this type of glia and further confirmed by the depressant effect of the nonspecific blocker of Piezo1 channels, Gd^3+^, on FM1-43- plus Yoda1-induced transient fluorescence peaks. Our proposal on the functional involvement of Piezo1-expressing SGCs in trigeminal nociception is consistent with the recent hypothesis that glial cells, apart from their supportive role, also functionally modulate neuronal activity [26,27,28]. Therefore, the SGCs surrounding the bodies of trigeminal neurons represent a new target for blocking the pro-nociceptive activity of Piezo channels in the trigeminovascular system. This complex system, presented by the trigeminal nerve and its projections to the meningeal tissues enriched by blood vessels, is supposed to be a key contributor to migraine pain [17,19]. In our study, not only isolated neurons and SGCs, but a plethora of cell types remained persistently labelled in the meninges after short-term exposure to FM1-43. However, one limitation of this study was that we did not establish the identity of these cell types, which can be addressed in further studies. We also cannot exclude similarities in the role of Piezo1 subunits compared to Piezo2 subunits, which are expressed in tactile innervation [14]. However, such studies are limited by the lack of specific chemical agonists for this subtype of mechanosensitive channel. 

In practical terms, the lasting labeling of living tissues by FM1-43 can be used for the non-invasive monitoring of Piezo1 activity in in vivo conditions. Based on our findings regarding the specific chemical activation of Piezo1 channels, we can hypothesize that Piezo1-activity-dependent labeling could also be induced either by physiological or disease-associated mechanical forces. One of the possible applications for migraine research would be an injection of FM1-43 followed by the triggering of a migraine attack with trinitroglycerin [29] or optogenetics/the KCl-induced phenomenon of cortical depolarizations as a model of migraine with aura [30,31]. Alternatively, FM1-43 could be co-administered with Yoda1, which has been shown to be safe and effective for in vivo administration [32]. Successful previous activations of Piezo receptors in the meninges or other parts of the trigeminal nociceptive system in vivo could then be detected postmortem.

In summary, our results demonstrate that the long-lasting labeling of trigeminal cells with FM1-43 supports previous proposals on the use of this dye for cell labelling via mechanosensitive channels [12]. However, that previous study did not test the nociceptive system and did not identify the molecular identity of respective membrane pathways. Here, we show the novel application of this methodological approach in a trigeminal culture model comprising of neurons and SGCs, both expressing Piezo1 channels as molecular targets mediating FM1-43 trapping. Moreover, we demonstrate the long-lasting memory of past Piezo1 activity in the ex vivo meningeal preparation, in which FM1-43 dye remained detectable several hours after free dye washout. This long-lasting FM1-43 “tail response” can be used in in vivo studies to explore the mechanosensitive Piezo1 mechanisms in physiological and disease models. Further investigations should estimate the maximal longevity of the “tail response” and outline a full spectrum of FM1-43 usability.

## 4. Materials and Methods

### 4.1. HEK293T Cultures and Transfection

HEK293T cells cultures were maintained as reported previously [33,34] in Dulbecco’s modified Eagle’s medium (DMEM, GibcoInvitrogen, Waltham, MA, USA) supplemented with 10% FBS and antibiotics. The cells were passaged at 80% confluence. The HEK cells were transfected in 35 mm petri dishes (2/3 of confluence) using FugeneHD [35]. The transfection efficiency was quantified at the nanodrop 48 h after transfection, and the experiment was performed if the efficiency was >50%. Transfection was performed with plasmid DNA encoding Piezo1 channels.

### 4.2. Animals

The Animal House of the university of Eastern Finland provided male and female C57BL/6J mice for this study. In order to test Yoda1 and HOS, the cell culture was prepared from trigeminal ganglia of P10-P13 mice. The FM1-43 labelling experiments were conducted on the cell cultures prepared from P10-P13 and P30 mice. The animals were housed under the following conditions: 12 h dark/light cycle, grouped housing, ad libitum access to food and water and an ambient temperature of 22 °C. All the experimental procedures performed in this study followed the rules of the European Community Council Directive of 22 September 2010 (2010/63/EEC). The Animal Care and Use Committee of the University of Eastern Finland approved all the experimental protocols (license EKS-002-2017).

### 4.3. Primary Trigeminal Cultures

The trigeminal cell culture was prepared according to a protocol previously described [15]. In brief, after decapitation, the skulls were cut sagittally. The trigeminal ganglions were dissected and incubated in an enzymatic cocktail (30 min in 0.5 mg/mL of trypsin, 1 mg/mL of collagenase, 0.2 mg/mL of DNase, all Sigma-Aldrich, St. Louis, MI, USA). The isolated cells were plated on glass coverslips covered with 0.2 mg/mL of poly-L-lysine (Sigma-Aldrich, St. Louis, MO, USA) and were cultured for 1–2 days in F12 medium (Gibco, Billings, MT, USA) at 37 °C, 5% CO_2_ before the experiment.

### 4.4. Solutions

An isotonic basic solution (IBS) was used, containing (in millimolar) 152 NaCl, 5 KCl, 10 HEPES, 10 glucose, 2.6 CaCl_2_, and 2.1 MgCl_2_, pH adjusted to 7.4 with NaOH, with an osmolarity of ~320 mOsm/kg. Stock solutions of Yoda1 were prepared in DMSO and stored at −20 °C at 50 mM for a maximum of one month. In order to test the selective activation of Piezo1, 5 μM of Yoda1 was applied. In order to test whether Yoda1 coupled to FM1-43, 2 μM of FM1-43 and 25 μM of Yoda1 were applied. The Gadolinium (Gd^3+^) stock, a nonselective Piezo1 channel blocker, was prepared in mQ water and stored at −20 °C at 50 mM. It was then applied in the experiment as 100 μM of Gadolinium. In the imaging experiments, we performed constant application of DMSO vehicle solution containing the same concentration of DMSO as the Yoda1 solution. Furthermore, the solutions of chemicals that were not dissolved in DMSO were dissolved in it in order to standardize the experiment.

### 4.5. Live Imaging

In the first experiments to observe the selective activation of Piezo1 channels by Yoda1, HOS, or US, primary trigeminal cells were incubated for 30 min and HEK293T for 20 min in 1× Fluo-4 AM (Fluo-4 Direct Calcium Assay Kit, Invitrogen, Waltham, MA, USA) fluorescent dye at 37 °C. Then, the cells were post-incubated (10 min at 37 °C and 10 min at room temperature) in IBS. Next, the cells were placed in a TILL photonics imaging system (TILL Photonics GmbH, Gräfelfing, Germany), where they were constantly perfused with IBS containing 1:10,000 DMSO (with a flow rate of 3 mL/min). Our perfusion system (Rapid Solution Changer RSC200, BioLogic Science Instruments, Grenoble, France) allowed the application of various solutions and fast (∼30 ms) exchange between them. The cells were imaged with a 10× objective using an Olympus IX-70 (Tokyo, Japan) equipped with a CCD camera (SensiCam, PCO imaging, Kelheim, Germany). In all the experiments, the sampling frequency was set to two frames per second. The excitation wavelength was set to 494 nm for Fluo-4 AM and 480 nm for FM1-43. The following protocols were used: (a) Calcium imaging. Loading with Fluo-4 AM. Solutions applied: 20 s of 5 μM of Yoda1, used as a chemical selective agonist for Piezo1 receptors [23]; (b) FM1-43 internalization. Solutions for HEK293T: two applications (1 min each) of 2 μM of FM1-43 interleaved with 2 min IBS + DMSO, followed by 1 min of 2 μM of FM1-43 + 25 μM of Yoda1 application. Solutions for trigeminal cultures: 1 min of 2 μM of FM1-43, 2 min of IBS + DMSO, 1 min of 2 μM of FM1-43 + 25 of μM Yoda1, 2 min of IBS + DMSO, 1 min of 2 μM of FM1-43 + 25μM of Yoda1 + 100 μM of Gd^3+^ (with 1 min IBS + DMSO + Gd^3+^ before and after). The data was recorded and post-processed using Live Acquisition and Offline Analysis software (Till Photonics GmbH, Gräfelfing, Germany).

### 4.6. Hemiskull Staining

For the FM1-43 tests in meningeal preparations, we used 5-week-old C57BL/6J male mice. Hemiskulls with preserved meninges were prepared as described previously [18,36]. Briefly, after decapitation, the skulls were cleaned from the skin, muscles, and connective tissues and were cut sagittally, resulting in two hemiskulls. From each pair, one hemiskull was used as a control (10 min in 10 µM of FM1-43), whereas the contralateral hemiskull was used for Yoda1 treatment (10 min in 10 µM of FM1-43 + 25 µM of Yoda1). The hemiskulls were imaged after 3.5 h of washout (in constantly oxygenated (5% CO_2_/95% O_2_) aCSF containing (in mM): 120 NaCl, 3 KCl, 2 CaCl_2_, 1 MgCl_2_, 1 NaH_2_PO_4_, 25 NaHCO_3_, and 10 glucose, pH 7.3). Imaging was conducted with an Axio Zoom microscope (Zeiss, Jena, Germany) at 4× magnification using ZEN 2 software (Zeiss, Jena, Germany).

### 4.7. Statistical Analysis

Data were analyzed and plotted using GraphPad Prizm (GraphPad Prizm Software, La Jolla, CA, USA) and Origin (OriginLab Corporation, Northampton, MA, USA). The data are presented as mean ± SEM (standard error of mean). Differences were considered statistically significant at *p* < 0.05. Student’s paired and unpaired t-tests and mixed effect analysis with Tuckey post hoc tests were used to detect statistical significance. The raw data supporting the conclusions of this manuscript will be made available by the authors, without undue reservation, to any qualified researcher.

## 5. Conclusions

In this study, for the first time, we demonstrated the long-lasting intracellular labeling of trigeminal neurons and a specific type of glial cells with the styryl FM1-43 dye, mediated by chemically stimulated mechanosensitive Piezo1 channels. This Piezo1-activity-dependent property of FM1-43, which appears to be non-toxic for living tissues, allows to combine both in vivo and in vitro studies of Piezo1-mediated mechanosensitive nociception in the trigeminovascular system. This discovery could help in the monitoring of both previous and ongoing activity of Piezo1 channels in the trigeminovascular system, the origin site of headaches in migraines.

## Figures and Tables

**Figure 1 ijms-24-01688-f001:**
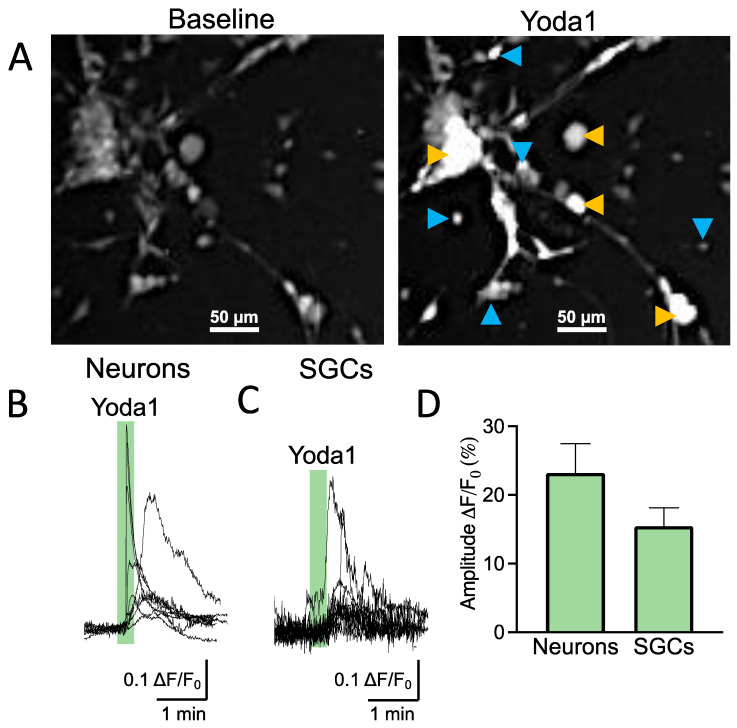
Live imaging shows that Piezo1 channels are functionally expressed on SGCs and neurons. (**A**). Fluorescent snapshots of 2DIV mouse trigeminal culture showing SGCs (blue arrows) and neurons (yellow arrows). The snapshots have been taken after 20 min of incubation with Fluo-4 AM. The left frame represents the trigeminal culture during baseline. The right frame shows cell activation after the application of 5 µM of Yoda1. Scale bar—50 μm. (**B**). Sample traces of calcium transients in trigeminal neurons induced by 5 µM of Yoda1 (*n* = 9). (**C**). Sample traces of calcium transients in trigeminal SGCs induced by 5 µM of Yoda1 (*n* = 12). (**D**). Statistics of the percentage of increased fluorescence intensity normalized to baseline in neurons (*n* = 9) vs. SGCs (*n* = 12) treated with 5 μM of Yoda1. There is no difference in Yoda1 responses among the different cell types (unpaired *t*-test, *p* = 0.120).

**Figure 2 ijms-24-01688-f002:**
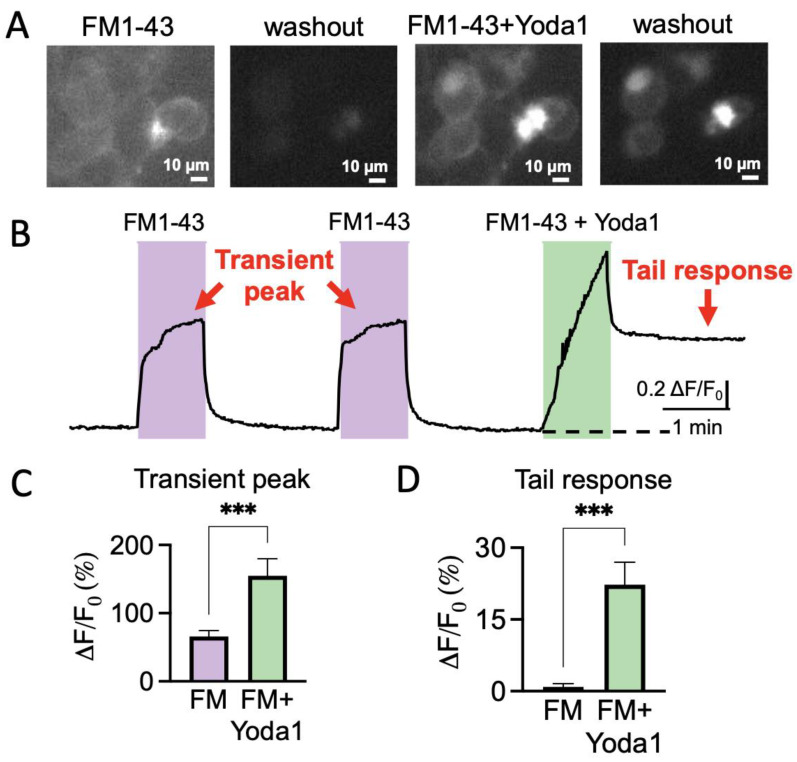
FM1-43 remains trapped after application of the Piezo1 agonist Yoda1 in Piezo1-transfected HEK cells. (**A**). Fluorescence snapshots of Piezo1-transfected HEK cells after 1 min application of 2 μM of FM1-43 (1st frame), during washout following only FM1-43 application (second frame). The third frame is captured at the top of 2 μM of FM1-43 + 25 μM of Yoda1 treatment (transient peak), followed by washout (fourth frame), where the dye remained trapped (tail response). Calibration bar—10 μm. (**B**). Sample trace of fluorescence intensity variations induced by double application of 2 μM of FM1-43 and following 2 μM of FM1-43 + 25 μM of Yoda1 for Piezo1-transfected HEK cells. Important elements in order to understand the outcome of the experiment are the transient peak, showing a treatment-induced acute response, and the tail response, addressing the long-lasting dye internalization. (**C**). Statistics of the percentage of increased fluorescence intensity of the transient peak increase in different cells when 2 μM of FM1-43 + 25 μM of Yoda1 are applied, compared to only 2 μM of FM1-43 (paired *t*-test, *** *p* < 0.001). (**D**). Statistics of the percentage of increased fluorescence intensity of the tail response (baseline) increase in different cells when 2 μM of FM1-43 + 25 μM of Yoda1 are applied, compared to only 2 μM of FM1-43 (paired *t*-test, *** *p* < 0.001).

**Figure 3 ijms-24-01688-f003:**
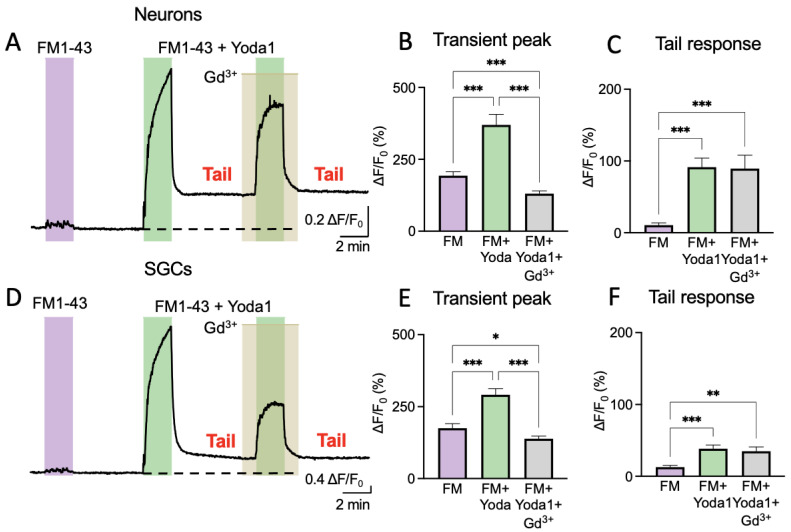
Trigeminal cell FM1-43 long-lasting labelling is mediated by Piezo1 activation. (**A**). Sample trace of fluorescence intensity variations in trigeminal neurons induced by 2 μM of FM1-43 followed by 2 μM of FM1-43 + 25 μM of Yoda1 and, in the end, 2 μM of FM1-43 + 25 μM of Yoda1 + 100 μM of Gadolinium (Gd^3+^). Important elements are the transient peak, showing the acute treatment response, and the tail response, addressing long-lasting dye internalization. (**B**). Percentage of increased fluorescence intensity in trigeminal neurons. The increase in the transient peak was normalized to baseline during the application of FM1-43 followed by 2 μM of FM1-43 + 25 μM of Yoda1 and, in the end, 2 μM of FM1-43 + 25 μM of Yoda1 + 100 μM of Gadolinium (Gd^3+^). The transient peak was higher only when both FM1-43 and Yoda1 were applied (mixed effect analysis with Tuckey post-hoc test, *** *p* < 0.001). (**C**). Percentage distribution of increased tail response levels in neurons normalized to the initial baseline after 2 μM of FM1-43 followed by 2 μM of FM1-43 + 25 μM of Yoda1 and, in the end, 2 μM of FM1-43 + 25 μM of Yoda1 + 100 μM of Gadolinium (Gd^3+^). The tail response was higher only when both FM1-43 and Yoda1 were applied (mixed effect analysis with Tuckey post-hoc test, *** *p* < 0.001). (**D**). Sample trace of fluorescence intensity variations in SGCs induced by 2 μM of FM1-43 followed by 2 μM of FM1-43 + 25 μM of Yoda1 and, in the end, 2 μM of FM1-43 + 25 μM of Yoda1 + 100 μM of Gadolinium (Gd^3+^). Important elements are the transient peak, showing the Yoda1 response, and the tail response, addressing long-lasting dye internalization. (**E**). Percentage of increased fluorescence intensity in SGCs. An increase in the transient peak was normalized to baseline during the application of 2 μM of FM1-43 followed by 2 μM of FM1-43 + 25 μM of Yoda1 and, in the end, 2 μM of FM1-43 + 25 μM of Yoda1 + 100 μM of Gadolinium (Gd^3+^). The transient peak was higher only when both FM1-43 and Yoda1 were applied (mixed effect analysis with Tuckey post-hoc test, * *p* = 0.037, *** *p* < 0.001). (**F**). Percentage distribution of increased tail response levels in SGCs normalized to the initial baseline after the application of 2 μM of FM1-43 followed by 2 μM of FM1-43 + 25 μM of Yoda1 and, in the end, 2 μM of FM1-43 + 25 μM of Yoda1 + 100 μM of Gadolinium (Gd^3+^). The tail response was higher only when both FM1-43 and Yoda1 were applied (mixed effect analysis with Tuckey post-hoc test, ** *p* = 0.001, *** *p* < 0.001).

**Figure 4 ijms-24-01688-f004:**
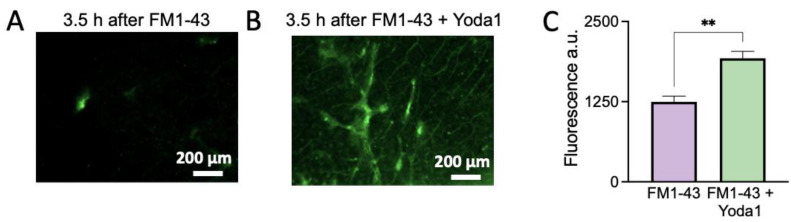
FM1-43 highlights previous Yoda1-induced Piezo1 activation in mouse ex vivo meninges. (**A**). Microphotographs demonstrating the fluorescence of meninges after 10 min of staining with only FM1-43 treatment as a control, followed by 3.5 h of washout. Few membranes are stained and visible. (**B**). Microphotographs demonstrating the fluorescence of meninges after 10 min if staining with FM1-43 + Yoda1, followed by 3.5 h of washout. Many meningeal structures are labelled. (**C**). Fluorescence of a whole imaged region was 1247 ± 90.2 a.u. in control (FM1-43) vs. 1925 ± 108.1 in FM1-43 + Yoda1 treated samples (*n* = 7, paired t-test, ** *p* = 0.002).

## Data Availability

The data presented in this study are available on request from the corresponding author.

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
