# Peer review of "FM1-43 Dye Memorizes Piezo1 Activation in the Trigeminal Nociceptive System Implicated in Migraine Pain"

_ijms, 2023, doi:10.3390/ijms24021688_

Round 1

Reviewer 1 Report

This paper demonstrated the long-lasting intracellular labeling of trigeminal neurons and specific type of glial cells with the styryl FM1-43 dye, monitoring ongoing and previous activation of Piezo1 channels in the trigeminal nociceptive system, implicated in migraine pain. Overall, the article is well organized and its presentation is good. However, some minor issues still need to be improved:

1. What is the reason for choosing 5µM in 84-85 lines but using 25µM in 86 lines?

2. Why the quantity is not consistent in Figure 1B-C?

3. Please add scale to the Figure 1A and Figure 2A.

4. Why the Transient peak is so different after using FM1-43 alone to label the cells in Figure 2C and Figure 3C?

5. The language is also hard to understand and should be polished.

Author Response

Reviewer 1

This paper demonstrated the long-lasting intracellular labeling of trigeminal neurons and specific type of glial cells with the styryl FM1-43 dye, monitoring ongoing and previous activation of Piezo1 channels in the trigeminal nociceptive system, implicated in migraine pain. Overall, the article is well organized and its presentation is good. However, some minor issues still need to be improved:

Reviewer 1: What is the reason for choosing 5 µM in 84-85 lines but using 25 µM in 86 lines?

Response: We thank the Reviewer 1 for noticing this typo at line 86. It has now been corrected. The Yoda1 concentration applied in this set of experiments was 5 mM (corrected in new line 90).

Reviewer 1: Why the quantity is not consistent in Figure 1B-C?

Response: In heterogenous trigeminal cultures, neurons are less in number than surrounding satellite glial cells (notice, Figure 1A). This feature is in line with general known ratio of neurons-glial cells. According to this legit question, we now pointed this out in line 69.

Reviewer 1: Please add scale to the Figure 1A and Figure 2A.

Response: Thanks for this suggestion. We have added scales to all images in Figure 1A (50 µm) and Figure 2A (10 µm).

Reviewer 1: Why the Transient peak is so different after using FM1-43 alone to label the cells in Figure 2C and Figure 3C?

Response: Figures 2 and 3 show the effect of FM1-43 in different cell types. Indeed, the membrane labelling of FM1-43 alone of transfected HEKT cells overexpressing Piezo1 channels appears to be more efficient than the one of trigeminal neurons and SGCs. In addition, it could be related to the different membrane properties, distribution, and composition of lipids (known to be efficient modulators of Piezo1 channels) in each cell type.

Reviewer 1: The language is also hard to understand and should be polished.

Response: We thank the Reviewer for this suggestion. Now the paper underwent language polishing and changes (labelled in red) are done at lines 31, 41, 50, 53, 69, 70, 96, 99, 119, 154, 159, 163, 248, 253, 256, 258-268, 271, 286-293.

Reviewer 2 Report

The authors' research is of undoubted scientific and clinical interest. The manuscript is well structured, informative, figures increase the visibility of the results obtained.

The manuscript needs a little technical revision: it is desirable to increase the number of keywords to increase the visibility of this article in databases; 12 links are older than 10 years (I recommend updating the links).

Author Response

Reviewer 2

The authors' research is of undoubted scientific and clinical interest. The manuscript is well structured, informative, figures increase the visibility of the results obtained.

Response: We would like to thank the Reviewer for finding our paper interesting

Reviewer 2: The manuscript needs a little technical revision: it is desirable to increase the number of keywords to increase the visibility of this article in databases; 12 links are older than 10 years (I recommend updating the links).

Response: Thank you for pointing out the chance to update our key words and reference list.

Now new keywords are: pain, migraine; nociception; Piezo1, trigeminal neuron, glia, FM1-43

Also, as suggested, older references have now been updated with more recent related papers and reviews (ref 1, 7, 8-9, 26-28).